# Does the Intensity of the Headache Differ According to the Level of Neck Disability in Chronic Migraine Patients?

**DOI:** 10.3390/ijerph192316307

**Published:** 2022-12-06

**Authors:** Dilara Onan, Paolo Martelletti

**Affiliations:** 1Spine Health Unit, Faculty of Physical Therapy and Rehabilitation, Hacettepe University, Ankara 06230, Turkey; 2Department of Clinical and Molecular Medicine, Sapienza University, 000189 Rome, Italy; 3Regional Referral Headache Centre, Sant’Andrea Hospital, 000189 Rome, Italy

**Keywords:** chronic migraine, headache, neck pain, neck disability level, neck disability index

## Abstract

Chronic migraine (CM) patients who report a high frequency and intensity of headaches also report neck pain (NP) and neck disability (ND) in neck activities that require stability. In this context, CM patients may report different headache intensities at different levels of ND. Our aim in this study is to investigate whether the intensity of headaches differs according to the level of ND in CM patients. Headache intensity and NP intensity were evaluated with the Visual Analog Scale (VAS), and ND was evaluated with the Neck Disability Index (NDI). A total of 142 patients who met the inclusion criteria were included in the study. The mean age was 53.24 ± 12.08 years. The median number of monthly headache days was 20. According to VAS, the median headache intensity was 10(4–10) cm and the median of NP intensity was 9(1–10) cm. The mean NDI was 28.45 ± 10.28. There was a difference in headache intensity between mild and severe disability levels (*p* = 0.007, Z = −3.289); headache intensity between mild and complete disability levels (*p* = 0.000, Z = −4.421); and headache intensity between moderate and complete disability levels (*p* = 0.004, Z = −2.212). Although the difference in headache intensity between ND levels is small, a median increase of 2 cm in headache intensity at mild ND levels may result in complete ND. A median increase of 1 cm in headache intensity at the moderate ND level may cause complete disability in the neck. According to our results, the intensity of headaches of CM patients differed according to the level of ND. We consider our results to be clinically important in this context.

## 1. Introduction

Migraine is a disease that progresses with attacks and may be accompanied by headaches, vomiting, nausea, phonophobia, or photophobia. Chronic migraine is defined as a headache with migraine features on at least 15 days or more per month, for more than three months, and on at least 8 days of the headache days [1]. The result of these accompanying problems and the chronicity of the migraine can lead to disability and reduced quality of life [2,3].

Neck pain, one of the problems that can be seen together with migraine, is one of the most common complaints of patients [4,5]. One mechanism considered in the relationship between migraine and neck pain is that nociceptive stimulation from the dura mater and cervical musculoskeletal structures combines with secondary neurons in the trigeminocervical complex, and causes activation as a result of long-term stimulation of the caudal nucleus of the trigeminal nerve [6,7,8,9]. Neck pain is twice as common in chronic migraine as in episodic migraine, and neck disability is associated with chronic migraine [10]. According to the Global Burden Disease study 2019, headache disorders are ranked 3rd for both sexes and all ages, and 1st for both genders and ages 15–49 [11]. In a 2010 study, it was stated that neck pain ranks 4th among the diseases that cause global disability [12].

In a meta-analysis published in 2022, it was recommended to conduct studies with tools such as the Neck Disability Index (NDI), which evaluates disability in order to better define the problem of neck pain in migraine patients [5]. The NDI assesses neck pain intensity, self-care, lifting, reading, headache intensity, concentration, work, driving, sleep, and leisure activities [13]. As the deterioration in these activities, which affect daily life and include postural arrangements, increases, the level of disability also increases [10]. In Florencio’s study on chronic migraine patients, when the parameters evaluated in the NDI were listed, it was found that headache was the first among the answers with the highest disability, followed by the activity of reading and lifting [10]. On the other hand, considering the parameters it includes, the NDI score is a complex evaluator of migraine disability and neck disability affected by hypersensitivity [14]. Although literature studies have reported the relationship between neck pain and disability in chronic migraine patients, it has been stated that it is not known whether the reason behind the neck disability is a local muscle disorder due to migraine-induced disuse or impaired motor cortical origin [15]. Causes related to neck disability in chronic migraine have been investigated, it has been reported that the cervical region has less mobility without differences in muscle activity [16,17,18], neck movement speed, and craniocervical test endurance performance are lower [16,19]. However, some comments have reported that craniocervical test performance does not adequately reflect neck disability in chronic migraine [20]. Therefore, it is a matter of curiosity to investigate the neck-related complaints of chronic migraine patients.

When headache and neck disability are considered in chronic migraine patients, neck disability levels may vary due to migraine-induced disuse of the neck and head or pain inhibition. Although there are uncertainties in the causes of neck disability, it is known that the frequency of headache is related to neck disability [18]. Therefore, the intensity of headache may vary depending on the level of neck disability. As far as we know, there is no study investigating whether the intensity of headache differs according to the level of neck disability. The literature has made possible causes and general NDI assessments. Therefore, our aim in this study is to investigate whether the intensity of headache varies according to the level of neck disability in chronic migraine patients.

## 2. Materials and Methods

### 2.1. Study Design

This study is a retrospective open-label real-world study with ethical approval from Sapienza University (CE 5773_2020). The study was conducted between September and November 2022 in Sapienza University Sant’Andrea Hospital Headache Center in patients diagnosed with chronic migraine, by an internist with headache expertise according to the International Classification of Headache Disorders criteria 3rd edition [1]. All patients gave consent for routine evaluations. The inclusion criteria of the study were to be diagnosed with chronic migraine and to be between the ages of 18–65. The use of any painkiller or specific migraine medication was permitted as needed and was recorded in the Headache Diary. Exclusion criteria were the diagnosis of any accompanying headache other than chronic migraine, diagnosis of any pathology in the cervical spine, acute infection-fracture-inflammatory condition, and any systemic disease.

### 2.2. Outcome Measures

Neck disability level with Neck Disability Index (NDI), headache intensity, and neck pain intensity with Visual Analog Scale (VAS) was assessed. The age, body mass index, and monthly headache days of patients were recorded.

The NDI contains 10 items, each scored from 0 to 5, including neck pain, headache, weightlifting, reading, sleep, driving, work, self-care, concentration, and recreation (0–4 points = no disability, 5–14 points = mild, 14–24 points = moderate, 25–34 points = severe, 35 and above = complete disability) [13,21]. Headache and neck pain intensity was evaluated using the VAS. Patients selected their pain levels by marking them on a horizontal line between 0–10 cm (0 = no pain, 10 = very severe pain) [22]. The number of monthly headache days was recorded from the monthly headache diaries of the patients.

### 2.3. Statistical Analysis

The IBM SPSS statistical 23.0 software was used for the analyses. The probability plots, histograms, and the Kolmogorov-Smirnov or Shapiro-Wilk’s test of normality analyzed whether the numerical data were normally distributed. The results were presented as mean and standard deviation (SD) or median and minimum-maximum for continuous variables; number (n) and percentage (%) were given for categorical variables. The Kruskal–Wallis analysis was performed to compare headache intensity according to the level of neck disability (mild, moderate, severe, and complete). Statistical significance was accepted as *p* < 0.05 and The Bonferroni adjustment was used for the *p* value (0.05/4). The power analysis of the study was performed using the one-way ANOVA test. Alpha was 0.05, the effect size f-value was calculated as 0.37 and the power was found to be 0.97. Then, since the Kruskal–Wallis test is a non-parametric test, as suggested in the article by Prajapati et al., the Asymptotic Relative Efficiency (ARE) coefficient was multiplied by the power value found in the ANOVA test (0.955), and the power of the study for the Kruskal–Wallis test was found to be 0.926. The sample size (n = 142) was sufficient [23].

## 3. Results

A total of 142 patients who met the inclusion criteria. 90.1% of the sample were women. The mean age of the patients was 53.24 ± 12.08 years. The median migraine diagnosis year was 20. None of the patients studied were in the diagnostic range of Medication Overuse, according to the criteria of ICHD-3. The demographic and clinic information are presented in Table 1.

According to neck disability levels, 17 patients were at a mild level, 29 patients were at a moderate level, 55 patients were at a severe level, and 41 patients were at a complete disability level. All neck disability groups had a high headache intensity (Table 2). A Kruskal–Wallis test showed that headache intensity differed according to neck disability levels H(3) = 24.75, *p* = 0.000 (Table 2). Considering the disability levels, the headache intensity was the least at the mild neck level (*Mdn* = 8) compared to the other levels, while the headache intensity increased at the moderate neck level (*Mdn* = 9), and was at the highest neck level at the severe and complete neck levels (*Mdn* = 10) (Table 2). Post-hoc tests using an alpha level of 0.0125 (0.05/4) corrected for Bonferroni were used to compare all disability levels. There was a difference in headache intensity between mild and severe disability levels (*p* = 0.007, Z = −3.289). There was a difference in headache intensity between mild and complete disability levels (*p* = 0.000, Z = −4.421). There was a difference in headache intensity between moderate and complete disability levels (*p* = 0.004, Z = −2.212). None of the comparisons between other neck disability levels were significant after Bonferroni correction (all *p* > 0.0125) (Table 3). As a matter of fact, although the difference in headache intensity between the levels is small, an increase of 2 cm (as median) in headache intensity at a mild neck disability level may cause complete disability for the neck.

## 4. Discussion

This study investigated the intensity of headache in chronic migraine patients with different levels of neck disability. Headache intensity differed in chronic migraine patients according to neck disability levels. According to the results of post-hoc analysis, differences were determined between mild and severe disability levels, between mild and complete disability levels, and between moderate and complete disability levels.

When the spine is considered with all the muscles, joints, ligaments, and dura mater structures from the head to the coccyx, for example, as a result of the stimulation of the trigeminocervical junction, the neck region, which is the continuation of the chain, may also be affected by pain. There may be pain or movement limitations in this region, and neck stiffness may be reported by patients [24,25,26]. Stability and strength are required to sustain neck-related activity [10,27]. Considering the parameters evaluated in the neck disability questionnaire, weightlifting, reading, sleep, driving, work, self-care, and recreation activities require the strength and stability of the neck [13]. Therefore, muscle dysfunction is expected to be associated with chronic pain and disability [28].

Carvalho et al. evaluated chronic migraine patients in terms of neck disability and reported 28.8% no disability, 44.8% mild, 21.6% moderate, and 4.8% severe disability [29]. In the study of Rodriguez et al., 68 migraine patients reported neck pain. It was stated that 62 of these patients had neck disability and the median headache score was 8 points [20]. Neta et al. showed less mobility in the neck extensor muscle chain in chronic migraine patients compared to healthy female controls without headache (*p* < 0.05) [30]. In addition, when NDI levels were examined in chronic migraine patients, it was reported as 5% for none, 11% for mild, 19% for moderate, 30% for severe, and 33% for complete disability level [30]. In the study by Tolentino et al. in chronic migraine patients, it was stated that neck muscle strength may contribute to headache and neck pain (*p* < 0.05). In this study, the rates of patients by NDI level were reported as 19% for none, 51% for mild, 24% for moderate, and 6% for severe, and high neck disability indicates lower neck muscle strength (*p* < 0.05) [31]. Gil-Martinez et al. stated that neck disability was a predictive factor for headache for chronic migraine patients, and reported that the relationship between the neck disability score (NDI = 21 ± 8.8) and headache intensity was significant (r = 0.594, *p* < 0.001) [32]. Chronic migraine patients tend to report severe disability [33]. Carvalho et al., in their study, stated that the risk between severe neck disability and chronic migraine was 7.6 times and reported that chronic migraine patients had high disability scores and moderate-severe disability levels (*p* = 0.01) [33]. Florencio et al. detected neck disability in 92% of 65 chronic migraine patients (*p* < 0.001), and the mean headache intensity of the patients was 8.5 ± 1.5. 8% of the patients showed none, 35% mild, 31% moderate, 23% severe, and 3% complete disability level. In this study, it was emphasized that the risk of increase to mild (RR = 2.5), moderate (RR = 3.7), and severe (RR = 5.1) levels are significant (*p* < 0.05) [10]. The findings in our study support the literature results in terms of neck disability levels and headache intensity. The NDI includes 5 different levels, but in our results, none of the patients were at the no disability level. All patients had at least mild disabilities. In addition, when neck disability levels and headache intensity were examined, the mean NDI for mild disability (11.97%) was 10.58 ± 3.1, and the median for headache intensity was 8 cm; the mean NDI for moderate disability (20.42%) was 19.62 ± 3.55, and the median for headache intensity was 9 cm; the mean NDI for severe disability (38.73%) was 29.78 ± 2.55, and median for headache intensity was 10 cm; and the mean NDI for complete disability (28.87%) was 40.09 ± 4.75, and the median for headache intensity was 10 cm. Therefore, while the difference between the scores in NDI levels was high on average, the headache intensity of the patients was also high. These results indicate that in addition to the headache assessments of chronic migraine patients evaluated in the clinic, neck problems should also be evaluated. In chronic migraine patients with mild or moderate neck disability, a median increase of 1 or 2 cm in headache intensity may increase neck disability to severe or complete disability levels. It is also stated that neck disability in chronic migraine patients may be related to decreased mobility [17,18,19], and muscle strength-endurance [19] in the cervical region, while neck pain may develop due to persistent headache in migraine, due to non-use and to prevent further pain [15,16]. It is thought that these different results and interpretations may also be due to the heterogeneity of patient populations [29].

In light of the results of the literature studies and our findings, we think that evaluation and treatment methods may need to be shaped differently. In addition to drug therapy, patients can be given patient and pain education [34], a biopsychosocial model can be applied [35] or physical therapy and rehabilitation practices can be recommended [36]. Rezaeian et al. evaluated migraine patients in two groups as treatment and control groups and used myofascial release and stretching methods in the treatment group. In the treatment group, the NDI score decreased from 22.73 ± 7.12 to 9.42 ± 3.7 points (*p* < 0.001) [36]. Therefore, rehabilitation methods can be tried to reduce neck disability. On the other hand, Pinhero et al. suggested that research should be conducted during functional activities to understand the effect of cervical motor control on daily functional activities in migraine patients [16]. In general, we think that the findings of our study are of clinical relevance for further evaluation or treatment studies.

Our study has some limitations. First, due to the inherent nature of retrospective studies, and due to the fact that the results are not prospective, it is not possible to add evaluations for different time periods and therefore the results cannot be interpreted for different time periods [37]. On the other hand, as there was no control group in our study, we could not compare our results with episodic migraine patients or healthy controls. In addition, the generalizability of our study, which we conducted in a clinic where patients with headache-related problems applied, may be limited. In future studies, this situation can be generalized with different clinics and sample sizes. Finally, while NDI evaluates activities that require strength and stability, it also evaluates concentration and pain parameters, which can be affected by psychological, social, and work conditions [35,38]. Since our study was a retrospective study, we could not evaluate psychological, social, and work conditions. Therefore, it should be considered that pain and disability may be affected by the biopsychosocial situation.

## 5. Conclusions

Individuals with chronic migraine are prone to problems such as neck pain and disability. Patients with a mild or moderate level of neck disability may experience severe and complete neck disability with a 1 or 2-cm increase in headache according to VAS. So far, many literature studies have investigated the relationship between chronic migraine and neck problems, and neck mobility or muscle assessments that may cause it. It is understood that the results are generally at moderate-to-severe levels of neck disability, but there is no clear study reporting how much the intensity of headache changes according to neck disability. In our study, results were found that can present researchers and clinicians with the relationship between neck disability level and headache intensity as objective numerical data. In this respect, we think that our study contains clinically decisive and new guiding results in patient evaluations.

## Figures and Tables

**Table 1 ijerph-19-16307-t001:** The Demographic and Clinic Information.

Variables	Mean (SD) or Median (Min–Max) or n (%)
Age (years)	53.24 ± 12.08
Sex	
Female Male	128 (90.1%) 14 (9.9%)
BMI (kg/m^2^)	23.9 (14.70–36.60)
Headache Diagnosis (years)	20 (2–48)
Monthly Headache Days	20 (4–30)
Headache Intensity (VAS)	10 (4–10)
Neck Pain Intensity (VAS)	9 (1–10)
Neck Disability Level (NDI)	28.45 ± 10.28

BMI: Body Mass Index, SD: Standard Deviation, VAS: Visual Analog Scale, NDI: Neck Disability Index.

**Table 2 ijerph-19-16307-t002:** The Kruskal–Wallis Test Results and The Headache Intensity and Disability Scores According to Neck Disability Levels.

NDI Levels	n (%)	Mean Rank	X^2^	*p*	Mean (SD)/ Median (Min–Max)	
					(NDI)	(VAS)
Mild	17 (11.97)	40.97	24.75	0.000	10.58 ± 3.1/ 11 (6–14)	8(5–10)/ 8.12 ± 1.45
Moderate	29 (20.42)	58.22			19.62 ± 3.55/ 20 (12–24)	9 (6–10)/ 8.76 ± 1.21
Severe	55 (38.73)	74.75			29.78 ± 2.55/ 30 (25–34)	10 (4–10)/ 9.24 ± 1.15
Complete	41 (28.87)	89.2			40.09 ± 4.75/ 38 (33–50)	10 (5–10)/ 9.59 ± 1.0

VAS: Visual Analog Scale, NDI: Neck Disability Index, SD: Standard Deviation.

**Table 3 ijerph-19-16307-t003:** The Post-Hoc Test Results.

NDI Levels	Test Statistics	Std Error	Std. Test Statistic	Sig.	Adj. Sig.
Mild-Moderate	−17.254	11.382	−1.516	0.130	0.777
Mild-Severe	−33.775	10.340	−3.266	0.001	0.007 *
Mild-Complete	−48.225	10.749	−4.486	0.000	0.000 *
Moderate-Severe	−16.521	8.551	−1.932	0.053	0.320
Moderate-Complete	−30.971	9.041	−3.425	0.001	0.004 *
Severe-Complete	−14.450	7.688	−1.879	0.060	0.361

NDI: Neck Disability Index, * *p* < 0.0125 according to Bonferroni-adjustment.

## Data Availability

Not applicable.

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
