# Peer review of "Does the Intensity of the Headache Differ According to the Level of Neck Disability in Chronic Migraine Patients?"

_ijerph, 2022, doi:10.3390/ijerph192316307_

Round 1

Reviewer 1 Report

Dear Authors,

I read impressive and short study with interest. I just have some suggestions.

The statistical method mustn't be mentioned in the abstract part. Example: line 14, 16,17

In the first paragraph of introduction part, the reference [1] repeated over and over (line 38 and 40). I suggest to add the reference below for line 40 to refer the reduce of quality of life. 

Kurt, A., & Turhan, B. (2022). Physiotherapy Management of Migraine Pain: Facial Proprioceptive Neuromuscular Facilitation Technique Versus Connective Tissue Massage. Journal of Craniofacial Surgery33(8), 2328-2332.

Line 71: diagnosed with whom? must be stated!

Also the type of migraine??

Power of the study must be stated in the statistical analysis part.

In the inclusion/exclusion criteria: Usage of any painkillers or any medication???

Good luck.

Author Response

Response to Reviewer 1 Comments

Point 1: The statistical method mustn't be mentioned in the abstract part. Example: line 14, 16, 17.

Response 1: The statistical method was deleted in the abstract part (Line 14, 16, 17).

Point 2: In the first paragraph of introduction part, the reference [1] repeated over and over (line 38 and 40). I suggest to add the reference below for line 40 to refer the reduce of quality of life.

Kurt, A., & Turhan, B. (2022). Physiotherapy Management of Migraine Pain: Facial Proprioceptive Neuromuscular Facilitation Technique Versus Connective Tissue Massage. Journal of Craniofacial Surgery33(8), 2328-2332.

Response 2: On your suggestion, I added a reference to line 40. I've also added a related article reference.

Kurt, A., & Turhan, B. (2022). Physiotherapy Management of Migraine Pain: Facial Proprioceptive Neuromuscular Facilitation Technique Versus Connective Tissue Massage. Journal of Craniofacial Surgery33(8), 2328-2332.

Goksan, Y., B., Acar, E., Sancak, B., Sayin, E., Yalinay-Dikmen, P., Ilgaz-Aydinlar, E. The role of metacognition, negative automatic thoughts and emotions in migraine-related disability among adult migraine patients. Psychol Health Med 2022, 1-13.

Point 3: Line 71: diagnosed with whom? must be stated!

Response 3: On line 71, I added the information of the person who made the diagnosis as by an internist with headache expertise.

Point 4: Also the type of migraine??

Response 4: We mentioned it in the content of the article as chronic migraine.

Point 5: Power of the study must be stated in the statistical analysis part.

Response 5: Power of the study was stated in the statistical analysis part. “Statistical significance was accepted as P < 0.05 and The Bonferroni adjustment was used for the P value (0.05/4). The power analysis of the study was performed using the one-way ANOVA test. Alpha was 0.05, the effect size f value was calculated as 0.37 and the power was found to be 0.97. Then, since the Kruskal Wallis test is a non-parametric test, as suggested in the article by Prajapati et al., the ARE (Asymptotic Relative Efficiency) coefficient was multiplied by the power value found in the ANOVA test (0.955), and the power of the study for the Kruskal Wallis test was found to be 0.926. The sample size (n=142) was sufficient. [23].”

Point 6: In the inclusion/exclusion criteria: Usage of any painkillers or any medication???

Response 6: The usage of painkillers or any medication was stated in the methods section.The use of any painkiller or specific migraine medication was permitted as needed and was recorded in the Headache Diary.”

In the results section statement have been inserted:

None of the patients studied were in the diagnostic range of Medication Overuse according to the criteria of ICHD-3.

Reviewer 2 Report

The novelty shall be justified by highlighting that the manuscript contains sufficient contributions to the new body of knowledge. Thus, the literature survey should be extended to 2022, and the reference numbers of the newly pertinent journal papers need to be more clearly introduced, one by one. Moreover, you are requested to discuss how your approach differs from the cited references, then, reinforce your article’s innovation, and to revise the Introduction and Conclusion of the article.   The relevant thesis should be revised in terms of language. There are some spelling errors. 

Author Response

Response to Reviewer 2 Comments

Point 1: The novelty shall be justified by highlighting that the manuscript contains sufficient contributions to the new body of knowledge. Thus, the literature survey should be extended to 2022, and the reference numbers of the newly pertinent journal papers need to be more clearly introduced, one by one. Moreover, you are requested to discuss how your approach differs from the cited references, then, reinforce your article’s innovation, and to revise the Introduction and Conclusion of the article.   The relevant thesis should be revised in terms of language. There are some spelling errors. 

Response 1: The literature was searched as (chronic migraine) AND (neck disability) AND (headache intensity). First, 14 results were obtained for keywords. Then (chronic migraine) AND (neck injury) literature was searched and 156 results were obtained. Since 2022, articles have been added to the introduction and discussion sections of the article. The language corrections have been revised.
